# Effect of the Application of Virtual Reality on Pain Reduction and Cerebral Blood Flow in Robot-Assisted Gait Training in Burn Patients

**DOI:** 10.3390/jcm11133762

**Published:** 2022-06-29

**Authors:** Seung Yeol Lee, Jeong Yeon Cha, Ji Won Yoo, Matheu Nazareno, Yoon Soo Cho, So Young Joo, Cheong Hoon Seo

**Affiliations:** 1Department of Physical Medicine and Rehabilitation, College of Medicine, Soonchunhyang University Hospital, Bucheon 14158, Korea; shouletz@gmail.com; 2Department of Rehabilitation Medicine, College of Medicine, Hangang Sacred Heart Hospital, Hallym University, Seoul 07247, Korea; ckwjddus0115@naver.com; 3Department of Internal Medicine, School of Medicine, University of Nevada, Las Vegas, Las Vegas, NV 89154, USA; ji.yoo@unlv.edu; 4School of Life Sciences, University of Nevada, Las Vegas, Las Vegas, NV 89154, USA; nazarm1@unlv.nevada.edu

**Keywords:** burn pain, virtual reality, robot-assisted gait training

## Abstract

Burn injuries and their treatment are extremely painful. This study aimed to determine whether virtual reality (VR) could reduce pain during robot-assisted gait training (RAGT) in burn patients by analyzing the cerebral blood flow (CBF) in the prefrontal cortex over time using functional near-infrared spectroscopy (fNIRS). The patients included in this study complained of a pain score ≥5 on a visual analog scale (VAS) during RAGT, which was performed 10 times for 2 weeks. Each session consisted of 15 min of VR application, with a 2-min break, and 15 min without VR. The average values of oxyhemoglobin and deoxyhemoglobin concentrations in the prefrontal cortex on fNIRS were calculated at four stages: temporal delay time with only RAGT, RAGT without VR, temporal delay time with RAGT and VR, and RAGT with VR. The pain scores and CBF were evaluated in sessions 1, 5, and 10 of the RAGT. The mean VAS pain scores were significantly lower (*p* < 0.05) in the experimental condition than in the control condition. Oxyhemoglobin in the prefrontal lobe significantly increased when RAGT was performed with VR. In conclusion, VR may be a strong nonpharmacological pain reduction technique for burn patients during physical therapy.

## 1. Introduction

As the survival rate after burn injuries has greatly improved, owing to the development of acute treatment, the importance of rehabilitation for complications related to burns has also increased. Joint contractures due to hypertrophic scarring and pain are the most common complications caused by burns. The pain experienced when moving the burnt limb can discourage patients from undergoing physical therapy. However, if rehabilitation is not performed due to pain, joint contractures can progress and permanent disability may occur in the burnt limb. Robotic therapy for musculoskeletal disorders is performed to allow intensive and repetitive training to increase the range of motion (ROM) and improve motor function [1]. While the clinical effectiveness of exoskeleton robot-assisted rehabilitation has been confirmed in burn patients [2,3,4], robotic therapy cannot be performed in patients who complain of severe pain during treatment. Therefore, there is a need for research on treatment methods that can reduce pain during robotic therapy.

The primary treatment for controlling burn pain is medication. Nonetheless, medicines alone cannot be used to completely relieve pain because of their possible systemic adverse effects [5]. Hence, nonpharmacological treatment methods, including mental imagery, biofeedback, enhanced control, and hypnosis, have been applied [5]. Virtual reality (VR) is a technology that allows participants to feel as if they are in a virtual environment. Recently, VR has been used in rehabilitation as a distraction intervention [6]. Recent studies have confirmed the positive effects of VR application on both pain intensity and pain anxiety during acute burn treatments and procedures that can cause pain [5,6,7,8,9,10]. While there are several reports on VR applications in burn patients [11], only a few studies have conducted objective evaluations of subjective pain reduction when VR is applied to burn patients.

The mechanism underlying the analgesic action of VR is not clear; nevertheless, it probably involves diverting attention away from the noxious stimulus that initiates pain perception. It is thought that the prefrontal cortex (PFC) plays a key role in modulating pain during an attentional task [12,13]. The changes in cerebral blood flow (CBF) and metabolism due to chronic pain and the effects of various treatment modalities have been objectively measured using near-infrared spectroscopy (NIRS) [14]. NIRS is a non-invasive neuromonitoring technique that can measure the oxyhemoglobin (HbO_2_) and deoxyhemoglobin (HbR) concentrations in the cerebral blood while the study participants are moving [15,16]. A recent study has confirmed that functional near-infrared spectroscopy (fNIRS) can objectively measure pain perception [17]. The present study aimed to determine whether VR application could effectively reduce pain during robot-assisted gait training (RAGT) in burn patients.

## 2. Materials and Methods

This study enrolled 33 adult patients with partial-to-full-thickness burns that had spontaneously healed or required skin grafting from the Department of Rehabilitation Medicine at Hangang Sacred Heart Hospital in Korea between June 2020 and July 2021. The present study was registered at ClinicalTrials.gov (identifier: NCT05004766). Additionally, this study was conducted in accordance with the principles embodied in the Declaration of Helsinki and was approved by the Ethics Committee of Hangang Sacred Heart Hospital (approval no.: 2021-013). Written informed consent was obtained from all study participants. Prior to the commencement of the study, all patients rated their most severe pain during RAGT as a score of ≥5 on a visual analog scale (VAS) of 0 to 10, in which 0 represented no pain at all and 10 indicated the worst pain. We included patients aged >18 years with a functional ambulation category score of ≤3 (Table 1). The exclusion criteria were as follows: patients with a history of brain injury, cognitive disorders before burn injury, a medical condition that could have affected the brain structure, problems with weight bearing due to fracture or inflammation, skin disorders that could be worsened by RAGT, and patients with severe pain who could not wear the robot. During the study, the medication dosage was not adjusted to exclude changes in cerebral hemodynamics caused by drugs such as gabapentin or morphine, which could affect brain activation [18].

SUBAR^®^ (CRETEM, Anyang-si, Korea) is a wearable robot with a footplate that assists the patients’ gait. During RAGT, the therapist performed rehabilitation by adjusting the speed, step length, and degree of knee flexion according to the participants’ motor function. SUBAR^®^ allows passive movements of the lower limbs according to the adjusted parameters.

The order in which training was administered was arranged in a block paradigm. Each patient participated in the VR condition, during which RAGT was performed. Each patient also participated in the control condition, during which the participant underwent RAGT with no distractions for the same amount of time spent doing therapy in VR. RAGT was performed 10 times for 2 weeks, from Monday to Friday, for 30 min. Only robotic training without VR was performed for 15 min, and VR and RAGT were performed simultaneously for 15 min (Figure 1).

Using the VR system during RAGT, auditory simulation was applied along with the image of walking on a forest road or coastal road at the same speed as the walking speed of the robot. The VR programs are composed of scenic beauty with the sounds of nature. Each program is a blend of scenes such as the ocean, desert, forest, flowers, waterfalls, and wildlife.

The degree of PFC activation was measured while wearing the fNIRS device (NIRSIT^®^; OBELAB Inc., Seoul, Korea) on the head during RAGT (Figure 2).

Measurements were performed using a wearable fNIRS, which was fastened to the head using elastic straps inside a plastic cap. The middle of the marker aligned with the middle of the eyes (Nasion in the 10–20 system), and the bottom line of the device was positioned just above the study participants’ eyebrows. This system utilized 24 laser sources (780/850 nm; maximum power under 1 mW) and 32 photo detectors to measure signals from the PFC area. The device had 48 channels with a 3 cm distance between the laser and the detector [15]. The detected signals were filtered using a low-pass (DCT 0.1 Hz) and a high-pass filter (DCT 0.005 Hz) to minimize ambient light noise and motion-dependent noise. Each cycle consisted of four periods of 60 s each (temporal delay time with RAGT, RAGT without VR, temporal delay time with VR, and RAGT with VR (Figure 3).

The degree of PFC activation using NIRS was measured both when VR was applied and when it was not applied. The HbO_2_ and HbR concentrations in each session were measured on days 1, 5, and 10.

Pain, the primary dependent variable, was measured during training. At the end of each RAGT with VR and RAGT with the control condition, the patients were requested to report the pain score during training on days 1, 5, and 10. All patients verbally rated their pain as a score of 0 to 10 on a VAS, in which 0 represented “no pain at all” and 10 indicated “worst pain”. The patients rated the following factors: (1) how much time they spent thinking about their pain and/or burn wound (endpoints labeled as 0 min, the entire time); (2) how unpleasant the training was (not at all unpleasant to most unpleasant); (3) how much their wound bothered them (not at all bothersome to most bothersome); (4) their worst pain (no pain to worst pain); and (5) their average pain (no pain or worst pain).

Statistical analysis was performed using SPSS version 23 (IBM Corp., Armonk, NY, USA). The values were presented as mean ± standard deviation. Inter-condition scores (the condition with VR and the condition without VR) were compared using the Wilcoxon signed-rank sum test after conducting a normality test, with the significance level set at *p* < 0.05.

## 3. Results

On each day, the patients rated pain on the VAS during RAGT for each condition (once after RAGT with VR and once after the control condition). Except for the mean VAS pain rating of botheration (*p* = 0.12), the mean VAS pain ratings (time spent thinking about pain, unpleasantness, worst pain, and average pain) were significantly higher in the control condition than during VR application on day 1 (*p* < 0.001, *p* < 0.001, *p* < 0.001, and *p* < 0.001, respectively) (Table 2). The mean VAS pain ratings (time spent thinking about pain, unpleasantness, botheration, worst pain, and average pain) were significantly higher in the control condition than during VR application on day 5 (*p* < 0.001, *p* < 0.001, *p* < 0.001, *p* < 0.001, and *p* < 0.001, respectively) and day 10 (*p* < 0.001, *p* < 0.001, *p* < 0.001, *p* < 0.001, and *p* < 0.001, respectively).

The results of the analyses conducted on HbO_2_ in the PFC indicated a significant VR-related PFC activation during RAGT, as compared with the results in the control condition on day 1 (*p* = 0.03), day 5 (*p* = 0.03), and day 10 (*p* = 0.02) (Table 3 and Figure 4). The analysis of HbR in the PFC showed no significant differences between VR application and control conditions on day 1 (*p* = 0.45), day 5 (*p* = 0.77), and day 10 (*p* = 0.18) (Table 3).

## 4. Discussion

The findings of this study confirmed that VR application during RAGT significantly reduced the pain during training. In this study, the sensory (worst pain and average pain) and affective (unpleasant and bothersome) components of pain were evaluated. The time spent thinking about pain means procedural burn pain. VR reduced the patients’ pain scores for both sensory pain (ratings of worst and average pain) and affective pain (ratings of unpleasantness and botheration). In addition to the pain reduction described by the patients, increased PFC activation, which is the mechanism for the pain reduction with VR, was confirmed.

Pain includes multiple dimensions—sensory, affective, and cognitive. The findings of pain studies suggest that instead of an isolated dimension being involved in pain, it is a network of several interconnected brain dimensions. These brain regions comprise sensory (somatosensory cortex and insula), affective (insula and anterior cingulate cortex), and cognitive (prefrontal cortex [PFC]) dimensions. The PFC is involved in the memory of pain experience, emotion, and cognition. Several studies have shown that the PFC is associated with painful stimulation [15,19]. The PFC is activated when an individual consciously tries to suppress pain. The PFC activation resulting from an increased cognitive load during an attentional task, may inhibit the pain network, leading to diminished pain perception [13].

There is sufficient evidence regarding changes in the sensory and motor areas of the brain in patients experiencing pain. Modulation of these sensory and motor areas has been proven to be effective for pain control [20]. Reducing attention to pain can decrease pain perception. VR training is known to reduce pain by inducing attention to the VR environment [6,21]. Directing the attention to the VR environment as a result of audiovisual stimulation during training reduces pain awareness and increases pain tolerance [5]. Cognitive training with VR can interpret pain and modulate pain input to brain regions, reducing pain perception and pain-associated emotions [22]. Attentional distraction is an important mechanism that contributes to VR analgesia [14,23]. The mechanisms of attentional distraction involve changes in pain-inhibitory circuits [24]. The gate control theory may elucidate the pain reduction mechanism of VR [21,22,23]; that is, VR reduces pain perception by diverting attention away from pain [25]. Unlike other analgesics, which disrupt the C-fiber pathway that relays pain signals to the central nervous system, VR affects pain perception via attention and concentration [26]. Less attention to pain can result in a reduction in the amount of time spent thinking about pain [8,27]. With sensory-perception-motor response, VR has proven to be clinically useful even when applied in rehabilitation for patients with pain [28]. A previous study confirmed that pain is reduced when the motor ability is improved as a result of corticospinal tract stimulation [29]. With these mechanisms, robotic training and rehabilitation using VR have been shown to improve performance [30,31]. The decrease in mean pain ratings when applying VR in both the sensory and affective domains for pain observed in this study was the same as that observed in previous studies.

NIRS shows correlations with neural activities [16]. Brain activation has been confirmed to be highly correlated with HbO_2_. This phenomenon is reflected by the increase in blood flow to activated brain areas. With respect to the global increase in CBF, the observed hemodynamic change is mainly dominated by the sympathetic nervous system [32]. Pain exerts a substantially greater effect on HbO_2_ dynamics in the PFC and sensory–motor areas [15,17,33,34]. Given that changes in HbO_2_ are widely agreed upon as representing cortical activity, the results of several studies suggest that the pain experience is affected by the sensory–motor areas responsible for peripheral sensation and by the PFC, which is mainly responsible for the cognitive aspect of pain [14,35,36]. In this study, it was objectively confirmed by fNIRS that the activation of the PFC, a pain control area, was increased when VR was applied as compared with the control condition, in which only RAGT was performed.

Future studies should further expand the number and duration of virtual worlds used. Whether VR training that integrates multiple sensory and cognitive domains has any effect, as compared with simple distraction, warrants further investigation. Randomized controlled trials exploring the clinical effects of VR on parameters such as physical performance, ROM measurements, and pain during robotic therapy are required. Although this study comparatively analyzed only the degree of PFC activation using fNIRS, it is considered a great advantage in that it could be measured simultaneously during the experiment. The hemodynamic delay time was set from 15 s to <1 min in some randomized crossover trials [37,38]; however, the delay time of 1 min set in this study had the limitation that the “carried over” effect could not be completely excluded. Further randomized controlled trials are likely to be required in the future. Because this study was an open-label study, bias was possible. Nonetheless, the environment other than VR was not affected as much as possible during the study. In future research, we plan to simultaneously perform MRI to analyze pain-related subcortical areas.

## 5. Conclusions

This study confirmed that VR is an effective method for reducing pain when RAGT is applied to burn scars. Additional research on the value of VR analgesia during burn rehabilitation is warranted because of the potential of VR in new nonpharmacological techniques.

## Figures and Tables

**Figure 1 jcm-11-03762-f001:**
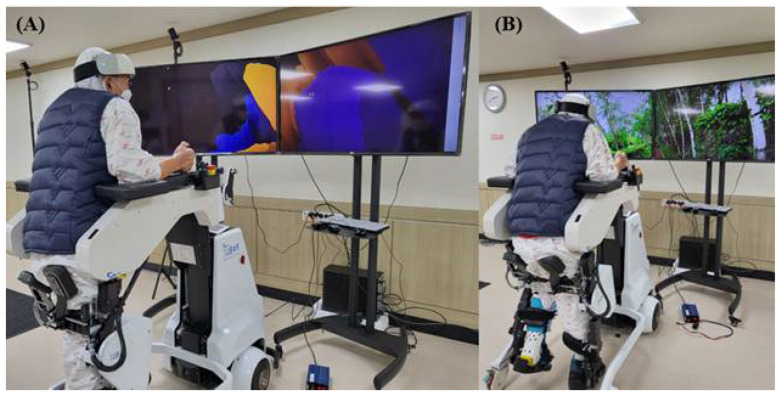
(**A**) Control condition without virtual reality. (**B**) The application of virtual reality to a patient during robot-assisted gait training.

**Figure 2 jcm-11-03762-f002:**
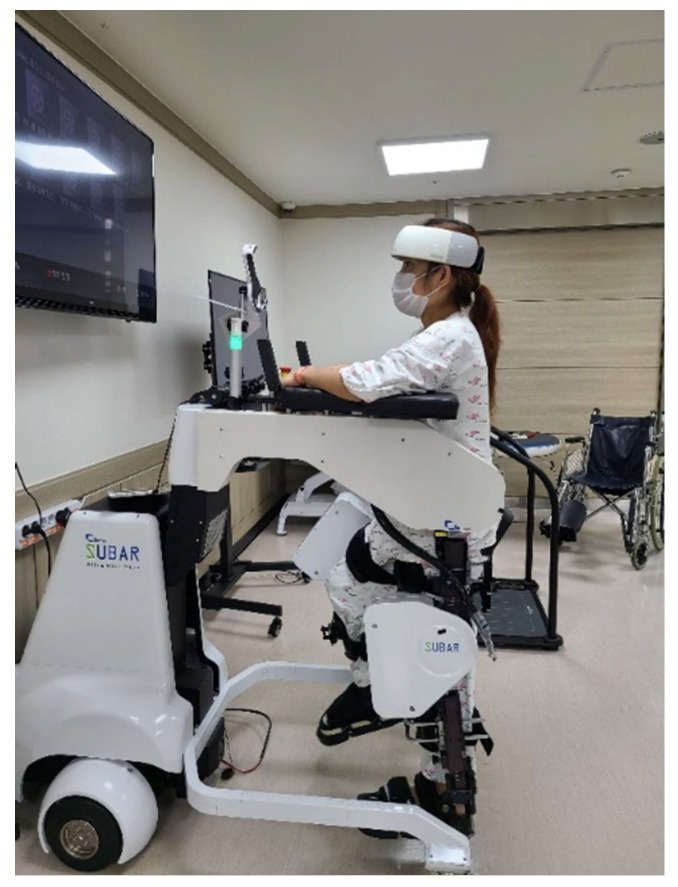
A patient wearing fNIRS during robot-assisted gait training.

**Figure 3 jcm-11-03762-f003:**
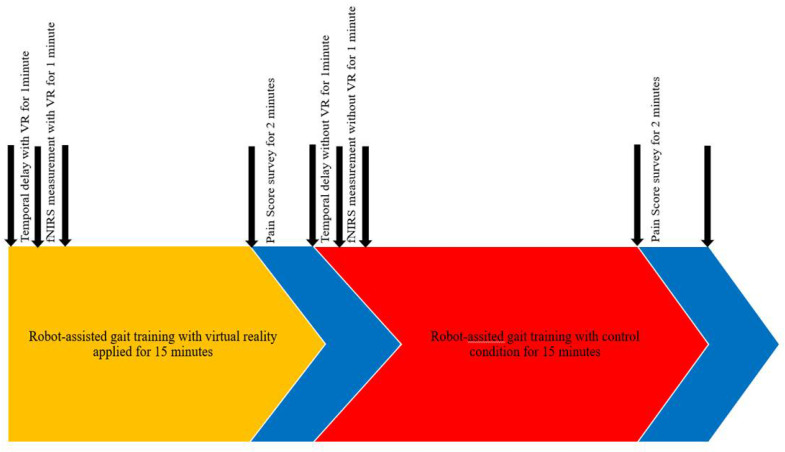
Block diagram of the protocol.

**Figure 4 jcm-11-03762-f004:**
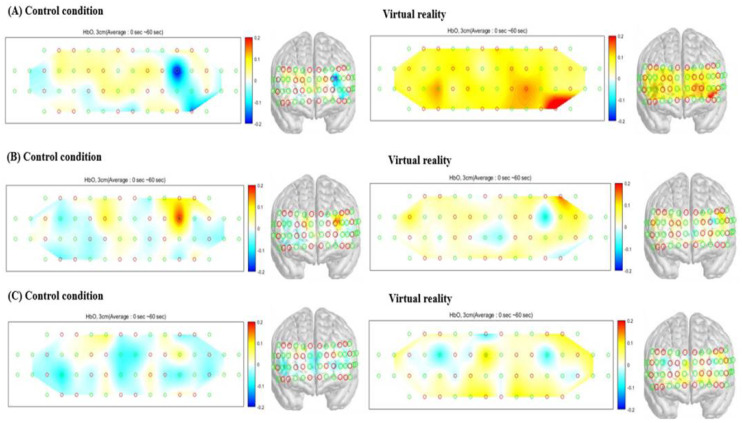
Reconstructed DOP images of HbO_2_. (**A**) Data from day 1, (**B**) data from day 5, and (**C**) data from day 10.

**Table 1 jcm-11-03762-t001:** Basic characteristics of the study participants.

	Participants (*n* = 33)
Male:female	27:6
Mean age (years)	57.55 ± 7.55
TBSA (%)	
Duration from burn to oxyhemoglobin level measurement (days)	106.82 ± 72.68
Mechanism of burn, *n* FB:EB:SB:CB	15:6:3:9
VAS	8.09 ± 1.01

TBSA, total body surface area; FB, flame burn; EB, electrical burn; SB, scaling burn; CB, contact burn; VAS, visual analog scale.

**Table 2 jcm-11-03762-t002:** Comparison of the mean pain scores between RAGT with VR and RAGT without VR.

	Day 1			Day 5			Day 10		
	Control	VR	*p*	Control	VR	*p*	Control	VR	*p*
Time spent thinking about pain	8.00 ± 1.79	6.09 ± 1.65	<0.001 *	7.45 ± 1.64	5.55 ± 1.75	<0.001 *	7.64 ± 1.39	4.91 ± 1.40	<0.001 *
Unpleasantness	8.00 ± 1.37	4.73 ± 1.89	<0.001 *	7.82 ± 1.42	4.55 ± 1.52	<0.001 *	7.36 ± 1.17	4.64 ± 1.69	<0.001 *
Botheration	8.09 ± 1.01	7.64 ± 1.39	0.12	7.82 ± 0.85	6.00 ± 1.37	<0.001 *	7.73 ± 1.07	4.82 ± 1.13	<0.001 *
Worst pain	8.64 ± 0.90	6.27 ± 1.63	<0.001 *	8.64 ± 0.90	6.18 ± 1.61	<0.001 *	8.10 ± 1.10	4.82 ± 1.36	<0.001 *
Average pain	8.00 ± 0.87	4.64 ± 1.75	<0.001 *	7.64 ± 0.49	4.36 ± 1.39	<0.001 *	7.73 ± 1.07	4.45 ± 1.58	<0.001 *

RAGT, robot-assisted gait training; VR, virtual reality. Values are presented as mean ± standard deviation. The *p*-values for between-condition differences were calculated using the Wilcoxon signed-rank sum test (*, *p* < 0.05), as appropriate.

**Table 3 jcm-11-03762-t003:** Comparison of HbO_2_ and HbR between RAGT with VR and RAGT without VR.

	Day 1			Day 5			Day 10		
	Control	VR	*p*	Control	VR	*p*	Control	VR	*p*
HbO_2_	0.00026 ± 0.00049	0.00055 ± 0.00071	0.03 *	0.00000 ± 0.00050	0.00043 ± 0.00072	0.03	−0.00020 ± 0.00067	0.00014 ± 0.00044	0.02 *
HbR	−0.00014 ± 0.00034	−0.00013 ± 0.00046	0.45	−0.00007 ± 0.00025	−0.00014 ± 0.00039	0.77	−0.0007 ± 0.00028	0.00003 ± 0.00026	0.18

HbO_2_, oxyhemoglobin; HbR, deoxyhemoglobin; RAGT, robot-assisted gait training; VR, virtual reality. Values are presented as mean ± standard deviation. The *p*-values for between-condition differences were calculated using the Wilcoxon signed-rank sum test (*, *p* < 0.05), as appropriate.

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
