# Peer review of "Effect of the Application of Virtual Reality on Pain Reduction and Cerebral Blood Flow in Robot-Assisted Gait Training in Burn Patients"

_jcm, 2022, doi:10.3390/jcm11133762_

Round 1

Reviewer 1 Report

The authors present a manuscript on Virtual Reality (VR) on pain reduction in robot-assisted gait training.  Additional data on brain blood flow are given. The authors performed a randomized, open label clinical study.

The manuscript confirms that VR is a non-pharmacological pain reduction technique. The authors present new information on near infrared spectroscopy to measures brain blood flow during VR giving a possible explanation for their results on VR.

While the manuscript confirms current knowledge on VR the study design and additional data on brain function warrants consideration.

Major concern:

At present the manuscript does not follow CONSORT criteria for reporting clinical trials. The authors should strictly follow these criteria which would  significantly improve their manuscript.

 2 minor  points should be  addressed:

There is a significant number of reports on VR in burn pain management.  The authors should at least cite  the metaanalysis on VR in burns (in line 55/56 of the manuscript).

 The authors  should discuss the impact of reduced pain levels on (objective) outcome e.g. ROM . What is known from the literature. The study design of the current study  limits a conclusion on better ROM after VR  and gait training . This should also be discussed. The authors should discuss a possible bias by an open label study.

Author Response

The authors present a manuscript on Virtual Reality (VR) on pain reduction in robot-assisted gait training.  Additional data on brain blood flow are given. The authors performed a randomized, open label clinical study.

The manuscript confirms that VR is a non-pharmacological pain reduction technique. The authors present new information on near infrared spectroscopy to measures brain blood flow during VR giving a possible explanation for their results on VR.

While the manuscript confirms current knowledge on VR the study design and additional data on brain function warrants consideration.

  1. Major concern. : At present the manuscript does not follow CONSORT criteria for reporting clinical trials. The authors should strictly follow these criteria which would  significantly improve their manuscript.

Answer> We appreciate you careful advise. Recently, NIRS has been used in several studies with the advantage of being able to measure brain activation at the same time during activity. In several studies, a period within 15 seconds to 1 minutes in a cross-over format is assumed and measured as hemodynamic delay time. We added references. We have added content to the discussion sections that we cannot completely rule out the “carried over” effect. We hope that these revisions will help readers understand more clearly.

  1. Minor  points should be  addressed: There is a significant number of reports on VR in burn pain management.  The authors should at least cite  the meta-analysis on VR in burns (in line 55/56 of the manuscript).

Answer> We appreciate you careful advise. We added references of the meta-analysis on the effects of VR on burn patients. With this reference, we were able to better understand what further research was needed in the future.

[10] Huang Q, Lin J, Han R, Peng C, Huang A. Using Virtual Reality Exposure Therapy in Pain Management: A Systematic Review and Meta-Analysis of Randomized Controlled Trials. Value in health : the journal of the International Society for Pharmacoeconomics and Outcomes Research. 2022;25:288-301.

[11] Czech O, Wrzeciono A, Batalík L, SzczepaÅ„ska-Gieracha J, Malicka I, Rutkowski S. Virtual reality intervention as a support method during wound care and rehabilitation after burns: A systematic review and meta-analysis. Complementary therapies in medicine. 2022;68:102837.

  1. The authors  should discuss the impact of reduced pain levels on (objective) outcome e.g. ROM . What is known from the literature. The study design of the current study  limits a conclusion on better ROM after VR  and gait training . This should also be discussed.

Answer> We appreciate you careful advise. Although this study did not confirm the correlation with objective data on final ROM improvement with randomized crossover trial, it was mentioned that research on physical improvement is also necessary by conducting randomised controlled trials in future studies.

  1. The authors should discuss a possible bias by an open label study.

Answer> We appreciate you careful advise. We added a limitation to the discussion section. We hope that these modifications will help readers better understand this study.

Reviewer 2 Report

This manuscript describes an interesting and novel approach to the management of pain during gait training in patients rehabilitating from burn injury using a distraction-style technique with virtual reality. The manuscript would benefit from copyediting to iron out some inconsistencies in the English. The flow of logic in the background section is clear, and it does a good job of setting up the need for the research – the hypothesis and objectives also clearly follow. The discussion is thorough and relevant. I generally think this could be an important contribution to the way we care for burn patients, though I am concerned primarily with the statistical analysis.

I think the manuscript would be greatly improved by following the CONSORT guidelines for reporting a randomized crossover design trial (Dwan, K., Li, T., Altman, D. G., & Elbourne, D. (2019). CONSORT 2010 statement: extension to randomised crossover trials. BMJ).

Pursuant to the above, I have some concerns with the statistical analysis: the study uses a crossover design, but does not mention assessing for carryover or period effects. The washout period between treatment is short (~2 minutes), which increases the risk of carryover effects; carryover effects should therefore be assessed and accounted for in the analysis. Similarly, there appears to be a strong period effect in the fNIRS data (the treatment effect of VR on HbO2 diminishes over time) – this should also be assessed and accounted for in the analysis. I think that the complexity of the experimental design calls for a mixed model analysis, though a more formal statistical opinion than mine may be warranted.

Author Response

  1. I think the manuscript would be greatly improved by following the CONSORT guidelines for reporting a randomized crossover design trial (Dwan, K., Li, T., Altman, D. G., & Elbourne, D. (2019). CONSORT 2010 statement: extension to randomised crossover trials. BMJ). Pursuant to the above, I have some concerns with the statistical analysis.

Answer> We appreciate you careful advise. We reviewed the reference you recommended. Therefore, we checked the checklist of randomised crossover trials and performed the Wilcoxon signed-rank sum test after normality test recommended by statistical analysis.

  1. The study uses a crossover design, but does not mention assessing for carryover or period effects. The washout period between treatment is short (~2 minutes), which increases the risk of carryover effects; carryover effects should therefore be assessed and accounted for in the analysis. Similarly, there appears to be a strong period effect in the fNIRS data (the treatment effect of VR on HbO2 diminishes over time) – this should also be assessed and accounted for in the analysis. I think that the complexity of the experimental design calls for a mixed model analysis, though a more formal statistical opinion than mine may be warranted.

Answer> We appreciate you careful advise. Recently, NIRS has been used in several studies with the advantage of being able to measure brain activation at the same time during activity. In several studies, a period within 15 seconds to 1 minutes in a crossover format is assumed and measured as hemodynamic delay time. We added references. We have added content to the discussion sections that we cannot completely rule out the “carried over” effect. We hope that these revisions will help readers understand more clearly.

[37] St George RJ, Jayakody O, Healey R, Breslin M, Hinder MR, Callisaya ML. Cognitive inhibition tasks interfere with dual-task walking and increase prefrontal cortical activity more than working memory tasks in young and older adultsi. Gait & posture. 2022;95:186-91.

[38] Kim J, Lee G, Lee J, Kim YH. Changes in Cortical Activity during Preferred and Fast Speed Walking under Single- and Dual-Tasks in the Young-Old and Old-Old Elderly. Brain sciences. 2021;11.

Round 2

Reviewer 2 Report

I am satisfied with the changes, thank you for your attention to my recommendations.